# A Muscle Load Feedback Application for Strength Training: A Proof-of-Concept Study

**DOI:** 10.3390/sports11090170

**Published:** 2023-09-05

**Authors:** Lisa Noteboom, Anouk Nijs, Peter J. Beek, Frans C. T. van der Helm, Marco J. M. Hoozemans

**Affiliations:** 1Department of Human Movement Sciences, Vrije Universiteit Amsterdam, Amsterdam Movement Sciences, 1081 BT Amsterdam, The Netherlands; a.nijs@vu.nl (A.N.); p.j.beek@vu.nl (P.J.B.); m.j.m.hoozemans@vu.nl (M.J.M.H.); 2Department of Biomechanical Engineering, Delft University of Technology, 2628 CD Delft, The Netherlands; f.c.t.vanderhelm@tudelft.nl

**Keywords:** strength training, resistance training, feedback, injury, muscles, overload, overtraining

## Abstract

Muscle overload injuries in strength training might be prevented by providing personalized feedback about muscle load during a workout. In the present study, a new muscle load feedback application, which monitors and visualizes the loading of specific muscle groups, was developed in collaboration with the fitness company Gymstory. The aim of the present study was to examine the effectiveness of this feedback application in managing muscle load balance, muscle load level, and muscle soreness, and to evaluate how its actual use was experienced. Thirty participants were randomly distributed into ‘control’, ‘partial feedback’, and ‘complete feedback’ groups and monitored for eight workouts using the automatic exercise tracking system of Gymstory. The control group received no feedback, while the partial feedback group received a visualization of their estimated cumulative muscle load after each exercise, and the participants in the complete feedback group received this visualization together with suggestions for the next exercise to target muscle groups that had not been loaded yet. Generalized estimation equations (GEEs) were used to compare muscle load balance and soreness, and a one-way ANOVA was used to compare user experience scores between groups. The complete feedback group showed a significantly better muscle load balance (β = −18.9; 95% CI [−29.3, −8.6]), adhered better to the load suggestion provided by the application (significant interactions), and had higher user experience scores for Attractiveness (*p* = 0.036), Stimulation (*p* = 0.031), and Novelty (*p* = 0.019) than the control group. No significant group differences were found for muscle soreness. Based on these results, it was concluded that personal feedback about muscle load in the form of a muscle body map in combination with exercise suggestions can effectively guide strength training practitioners towards certain load levels and more balanced cumulative muscle loads. This application has potential to be applied in strength training practice as a training tool and may help in preventing muscle overload.

## 1. Introduction

Participation in fitness and strength training activities is becoming increasingly popular and brings along various health benefits [1,2]. Nevertheless, sedentary lifestyles remain a serious health issue worldwide [3]. In times of growing sedentary behavior and related adverse health problems, fitness and strength training activities can play a major role in improving public health worldwide, because they are readily accessible [3]. However, on the downside, there is a health risk related to fitness and strength training activities in the form of musculoskeletal injuries. In 2019, fitness injuries accounted for 17% (930,000) of all sports injuries in the Netherlands, ranking this sports domain second behind soccer (22%) [4]. The challenge is to stimulate people to engage in fitness and strength training and reap all the associated health benefits while reducing the risk of fitness-related musculoskeletal injuries.

In line with the widely used injury prevention model of van Mechelen et al. [5], the first steps towards prevention include gaining knowledge regarding fitness-related injuries and their etiology. Few studies have investigated fitness-related injuries in the general population [4,6]. However, more studies have been conducted on fitness-related injuries in more competitive sub-groups of fitness, such as CrossFit, powerlifting, and weightlifting [7,8,9,10,11]. Although these sub-disciplines differ from general fitness and strength training because they focus on specific exercises only, similar loading principles and objectives apply. Therefore, these sub-disciplines can provide meaningful information regarding fitness-related injuries. Based on the combined results reported in those studies, 58–59% of the injuries involved the muscles and tendons [6,7]. Most injuries were mild to moderate in severity, resulting in relatively short (training) time losses [6,8,9]. Regarding the injury onset (acute or overuse), overuse injury percentages varied between 25 and 54% of all injuries [4,6,8,9,11], which can be considered high compared to other sports [4]. A systematic review [12] reported ‘overuse’ and the ‘frequent use of high loads’ as two of the five main causes of injury and musculoskeletal pain in fitness. Moreover, two studies reported that the main injury cause mentioned by fitness practitioners was overload [6,13]. Overload can result from excessive training volumes or insufficient recovery time. In one study, the specific types of overload causing injury that were mentioned by practitioners included ‘using too much weight’, ‘executing too many repetitions’, or ‘performing too many activities’ [6]. Based on these findings, it seems that a large part of the fitness injury problem concerns muscle injuries that are often caused by overload or overuse or at least are perceived as resulting from those causes. 

According to step three of van Mechelen et al.’s injury prevention model [5], a preventive measure is needed to prevent muscle overload. Currently, only 18% of all fitness participants train under the continuous supervision of a personal trainer or instructor [6]. Without expert supervision, strength training practitioners may not know the appropriate weights and number of repetitions to choose and may not be fully aware of which muscles they are training, which could lead to overloading of specific muscles. A potential solution might be to provide practitioners with feedback regarding their muscle load during their workout via a mobile application. However, to our knowledge, no such application has been developed, perhaps due to the technical difficulties involved in assessing muscle load, as well as various other issues that should be addressed. The first factor to consider is the modality of the feedback (e.g., visual, acoustic, or haptic). Visual feedback has been proven to be an effective means of performance enhancement in strength training [14] and is easy to provide via a mobile application. A second factor is the timing of the feedback, which can be concurrent, i.e., during the performance, or terminal, i.e., after the performance. Concurrent feedback may help with understanding a complex task, especially during the early phases of learning, whereas terminal feedback may lead to better learning in easy tasks or during later phases of learning [15]. A third factor is the content of the feedback, which must be relevant to the task and understandable to the user [15]. For the purpose of providing practitioners with feedback regarding their muscle load during their workout, we opted for visual, terminal feedback on muscle load after each exercise, personalized to their own capacity. This type of feedback could potentially help practitioners to avoid overloading their muscles, and instead load all muscles with an appropriate intensity that matches their capacity and results in achieving the desired training goal (e.g., hypertrophy). Additionally, apart from the perspective of injury prevention, practitioners might be more motivated to keep on exercising once they receive stimulating information during their workout in relation to their training goals [14]. 

In summary, muscle overload injuries form a problem in strength training and may be prevented by providing practitioners with feedback regarding their muscle load. As an initial step towards this goal, we developed and evaluated an innovative muscle load feedback application for strength training, with the overarching long-term aim of reducing injury risk by avoiding overload and overuse. In this study, we sought to determine if the feedback application in question can aid users to achieve a more balanced muscle load, by avoiding overloading certain muscles while underloading other muscles. A more balanced load may contribute to the prevention of overload injuries, which is an assumption that must be confirmed in future controlled trials with prospective longitudinal research designs. However, self-reported muscle soreness, which may be considered a precursor of muscle injuries [16], can be assessed in the short term. The primary aim of this proof-of-concept study was therefore to investigate if muscle load feedback can improve the muscle load balance, muscle load level, and muscle soreness of strength training practitioners, while the secondary aim was to evaluate the user experience and motivational effect of the application. In particular, it was hypothesized that muscle load feedback improves the muscle load balance and muscle soreness and has a positive effect on the user experience of strength training participants. If these hypotheses are confirmed, the feedback application may be further refined and validated.

## 2. Materials and Methods

### 2.1. Participants

The minimum sample size required to obtain an effect size of 0.25 with a power of 80% and a statistical significance level of 5% was calculated by G*Power 3 [17], assuming a mixed ANOVA design, since the literature regarding power analyses for Generalized Estimation Equations (GEEs), the statistical method adopted in the present study, was lacking. A minimum required sample size of 21 participants was found. To be certain of adequate power, 30 healthy participants (sex: 16 men, 14 women, age: 37 ± 14 years, strength training experience: 2.4 ± 2.4 years; mean ± standard deviation (SD)) were included in the present study. The inclusion criteria were a minimum age of 18 years and no musculoskeletal injuries at the start of the study. Recruitment was aimed at obtaining an as heterogenous sample as possible in terms of sex, age, and strength training experience, to achieve generalizable results. The participants were recruited through voluntary response sampling. The study was approved by the local ethics committee of the Faculty of Behavioural and Movement Sciences of the Vrije Universiteit Amsterdam (VCWE-2021-188). All participants provided written informed consent. 

### 2.2. Muscle Load Feedback Application

The feedback application was developed in collaboration with the fitness company Gymstory [18]. Gymstory provides an automated gym exercise tracking solution, based on a sensor and mobile app (Gymstory, Amsterdam, The Netherlands). With this solution, the performed exercises, lifted masses, and number of performed repetitions and sets can be tracked and registered for each individual strength training athlete. Since the purpose of the study was to provide muscle load feedback, we extended the Gymstory app with additional information and functionalities to be able to estimate personalized muscle loads. To this end, multiple components had to be implemented. First of all, although Gymstory tracks the training volume and intensity by registering the repetitions and masses, the maximum masses that participants can lift for specific exercises must be known to personalize this intensity. Therefore, the option to add the one-repetition max (1RM) per exercise was implemented. Second, estimating the muscle load for an exercise requires knowledge of which muscle groups are contributing to that movement and to what extent. These muscle contributions can be estimated based on functional anatomical knowledge. A database that described primary and secondary muscle contributions for multiple exercises (Appendix A) was implemented in the Gymstory app. Thirdly, an equation to estimate an athlete’s personalized muscle load, based on recorded repetitions and masses, measured 1RMs, and estimated muscle contributions was added (Appendix A). The equation is based on the assumption that when an exercise is performed for 3 sets of 10 repetitions at 70% of the 1RM, which corresponds to guidelines from the American College of Sports Medicine (ACSM) [19], the target load is achieved for a primary muscle, and half of the target load is achieved for a secondary muscle. More details and assumptions for the equation are presented in Appendix A. This equation was implemented in the Gymstory app. It must be noted that the muscle load calculations remain to be validated. For the present study, muscle load estimates obtained by using this equation were deemed sufficient to examine if the concept of the feedback application works. If the concept proves successful, the accuracy of the calculations could be improved. 

By employing the implemented components, a personalized muscle load could be estimated with the Gymstory app after each exercise. To provide clear and understandable feedback to the practitioners, the muscle load was visualized in the app as a colored muscle load body map (Figure 1A). This visualization represents the cumulative load that is updated after each exercise. At the start of a workout session, all muscle groups in the body map are white, and after each exercise, the colors of the corresponding primary and secondary muscle groups are updated, with light green indicating light to medium cumulative muscle load, dark green indicating the target muscle load, and yellow/orange/red indicating a potential risk of muscle overload. In addition to the muscle load body map, another functionality was added to the app, which provided practitioners with advice for subsequent exercises (including appropriate sets, repetitions, and weights) that targeted muscle groups that had not or had barely been loaded yet (Figure 1B). This feature rested on the assumption that a more balanced cumulative muscle load would prevent overloading of specific muscles and underloading of other muscles, considering a total body workout. 

### 2.3. Procedures

The study design was a matched experimental study with three groups. A flow diagram of the study design is shown in Figure 2. The participants were first matched by an optimization method that minimized the variations in sex, age, and strength training experience, and subsequently randomly allocated them to one of three groups: control group, ‘partial feedback’ group, and ‘complete feedback’ group. All participants started with two intake sessions with an experienced fitness instructor in which their one-repetition maximum (1RM) was estimated by a submaximal test protocol for 18 selected exercises [20]. Submaximal 1RM testing involves a safer protocol than maximal 1RM testing in which the participant performs a set of repetitions with a submaximal weight until failure. The number of correctly executed repetitions and the lifted weight are used to estimate the 1RM based on a regression equation [20]. Next, all participants performed eight strength training sessions for four consecutive weeks (two per week). In each session, the participants were instructed to perform 8 exercises of their choice from the 18 preselected exercises. All participants were instructed to attempt to perform a full-body workout, loading all muscles evenly. Performed exercises, sets, repetitions, and weights were automatically tracked by the Gymstory sensor. Based on this data in combination with the measured 1RMs and functional anatomical knowledge of primary and secondary muscle contributions per exercise, the cumulative load on each of the muscle groups for each participant was individually estimated using the developed equation (Appendix A). The three groups received the following feedback regarding their muscle load during the eight workouts:Control group: no feedback.Partial feedback: this group received a visualization of the estimated cumulative muscle load after each exercise via the Gymstory app in the form of the muscle load body map (Figure 1A).Complete feedback: this group also received the muscle load body map after each exercise, and additionally received a list with suggestions for subsequent exercises, targeting muscle groups that had not or had barely been loaded yet (Figure 1A,B).

The morning after each workout, all participants received an online questionnaire with questions regarding their muscle soreness (Appendix A) and were invited to score the soreness per muscle group by self-assessment on a visual analogue scale (VAS) (0–100 mm), with 0 indicating ‘no soreness’ and 100 indicating ‘maximum soreness’. The participants were asked not to perform any other sports activities in the 24 h before or on the same day after the strength workout, in order to avoid muscle soreness from other sports at the time of the questionnaire. Furthermore, after the last workout, the user experience questionnaire (UEQ) was added to the questionnaires [21,22] to examine how the practitioners experienced the application. The UEQ consists of 26 pairs of opposing terms, each belonging to one of six scales: Attractiveness, Perspicuity, Efficiency, Dependability, Stimulation, and Novelty. The answers were rated on a 7-point Likert scale ranging from −3 (completely agree with the term on the left) to +3 (completely agree with the term on the right). 

### 2.4. Statistical Analyses

A frequency analysis was performed to describe the characteristics of each group. After a workout, the total cumulative muscle load of a participant (calculated by the equation in Appendix A) was presented as an array, with a percentage between 0 and 100+ for each muscle group. From this array, the SD was calculated as an indicator of cumulative muscle load balance. A lower SD meant a more even load distribution over all muscles, while a higher SD meant that, compared to the mean loading of the muscles, certain muscle groups were loaded extensively while other muscle groups were not. In addition, the mean of the cumulative muscle load was calculated after each session. For the muscle soreness, the mean was calculated for each session. The SD of the cumulative muscle load was expressed as a percentage of the mean cumulative muscle load of that session for normalization (i.e., the coefficient of variation; CV). The statistical analyses were performed in IBM SPSS Statistics 25. The cumulative muscle load balance, mean cumulative muscle load, and mean muscle soreness were compared between the three groups and eight workout numbers using Generalized Estimation Equations (GEEs) with an exchangeable working correlation structure [23]. GEEs were chosen as the regression technique because this technique can deal with correlated data (repeated measurements) and missing data (missing workouts). In the initial GEE regression model, the categorical variables group (type of feedback), workout number, and the interaction between group and workout number were included as predictor variables. If the parameter estimates showed that none or only a very few of the (in total 14) group × workout number interaction combinations were significant, and no clear interaction effect could be observed from the corresponding figure, the interaction term was removed from the model. The parameter estimates and their 95% confidence intervals (CIs) for the final GEE models were reported for all outcome parameters, indicating that an a priori α level of 0.05 was used to determine statistical significance. In all regression analyses, the control group and the first workout were always taken as a reference for computing the contrasts. 

The UEQ scores for the six scales were compared between groups using a one-way ANOVA. Effect sizes *r* are reported, where *r* = 0.1 represents a small effect, *r* = 0.3 a medium effect, and *r* = 0.5 a large effect. In the case of a significant effect, post hoc analyses with Bonferroni correction were conducted. In addition, the scores were compared to the established benchmark values for the UEQ, which were constructed based on a dataset containing 9905 responses from 246 product evaluations [22]. This informs us not only about the differences in user experience between the different groups in our study, but also about how the user experience holds up compared to a large set of other products. 

## 3. Results

In total, 30 participants completed 191 workouts, implying that 49 sessions were missed, e.g., due to illness. All participants completed at least four workouts. The participant characteristics per group are summarized in Table 1. An ANOVA on these data confirmed that the matching and randomization procedure was successful and that there were no between-group differences in sex (*p* = 0.886), age (*p* = 0.934), or strength training experience (*p* = 0.963) (Table 1). 

### 3.1. Muscle Load

The group means of the normalized standard deviation of the cumulative muscle load, which was used as an indicator of muscle load balance, are presented for all workouts in Figure 3A. It can be observed from Figure 3 that, in general, the cumulative muscle load CV was lowest for the complete feedback group, followed by the partial feedback group and the control group, while the between-group differences remained reasonably constant over all workouts. The initial GEE model consisted of group, workout number, and their interaction as predictor variables of the outcome cumulative muscle load CV. This analysis revealed only one significant interaction term (from a total of 14 interaction terms between the categories of group and workout number), confirming that the differences between groups remained similar throughout the workouts. The interaction between group and workout number was therefore removed from the model and the final GEE model for cumulative muscle load CV (Table 2) showed that the cumulative muscle load CV was, for all workouts, significantly lower in the complete feedback group compared with the control group (β = −18.9; 95% CI [−29.3, −8.6]). The cumulative muscle load CV for the partial feedback group, although lower, was not significantly different from the control group (β = −6.8; 95% CI [−19.5, 6.0]). There were no significant differences in cumulative muscle load CVs between each of the workouts 2–8 and workout 1 (the reference workout), indicating that the cumulative muscle load CV did not change over time for each of the groups. These results indicated that a more balanced (estimated) muscle load was achieved with the complete feedback, but not with the partial feedback, compared with the control group.

In addition, it was examined whether the group, workout, or the interaction between group and workout influenced the mean cumulative muscle load. The corresponding regression analysis revealed multiple significant interactions (Table 2). In line with what can be observed in Figure 3B, these analyses showed that the mean cumulative muscle load increased significantly more from workout 1 to workout 6 in the control group than in the partial feedback group (β = −24.0, 95% CI [−47.9, −0.1]) and the complete feedback group (β = −24.3, 95% CI [−46.3, −2.2]). Also, between workouts 1 and 8, the cumulative muscle load increased significantly more in the control group than in the partial feedback group (β = −37.6, 95% CI [−62.4, −12.9]) and the complete feedback group (β = −26.3, 95% CI [−49.5, −3.0]). Figure 3B shows that while the mean cumulative muscle loads were similar in all groups at the initial workouts, the mean cumulative muscle load in the control group increased throughout the workouts, whereas the load in the feedback groups remained similar, explaining the observed significant interactions between group and workout numbers. 

### 3.2. Muscle Soreness

It was investigated if the mean muscle soreness was influenced by the group, workout number, or interaction between group and workout number (Figure 3C). Significant interactions between group and workout were not observed, and the final regression model without the interactions only showed a significantly lower mean soreness at workout 4 compared with workout 1 (β = −1.9, 95% CI [−3.5, −0.2]) for all groups (Table 2).

### 3.3. User Experience 

The group mean scores for the six scales of the User Experience Questionnaire (UEQ) after the last workout for all three groups are presented in Figure 4. In the background of the figure, the benchmark scores for the different scales are presented. As can be observed in this figure, the scores ranged between bad and above average for the control group and the partial feedback group, whereas the scores were above average to excellent for all scales for the complete feedback group.

The one-way ANOVA revealed a significant effect of group on the scores of all six scales (Table 3). Post hoc analyses with Bonferroni correction showed that the complete feedback group rated the app significantly better than the control group on Attractiveness (*p* = 0.036), Stimulation (*p* = 0.031), and Novelty (*p* = 0.019), and significantly higher than the partial feedback group on Dependability (*p* = 0.019), although this difference was also nearly significant for Perspicuity (*p* = 0.051), Efficiency (*p* = 0.051), and Novelty (*p* = 0.053). There were no significant differences between the control group and the partial feedback group.

## 4. Discussion

The primary aim of the present study was to investigate if the use of a newly conceived muscle load feedback application could effectively improve the cumulative muscle load balance, muscle load level, and muscle soreness balance in strength training practitioners during total body workouts. The results revealed that practitioners who received feedback in the form of a body map and exercise suggestions during their workout achieved a more balanced cumulative muscle load compared with practitioners who did not receive any feedback, as opposed to practitioners who only received the body map. Moreover, it was found that the mean cumulative muscle load was similar for all groups in the first workouts, but the mean load in the control group increased throughout the workouts while the load remained the same in the feedback groups. The mean muscle soreness was similar for all groups and all workouts, implying that no evidence was found that the feedback can decrease muscle soreness. 

The secondary aim of the study was to evaluate the subjective experience with the application. User experience, as assessed using the UEQ and compared with the benchmark scores, showed that the complete feedback group rated the application good to excellent on all scales. Comparisons between groups showed that the rated scores were better for the complete feedback group than the other two groups on all scales. 

The feedback provided in the form of a body map and exercise suggestion effectively improved the cumulative muscle load balance of the participants, indicating that participants adhered to the feedback provided by the app. To verify that the feedback was used, participants were questioned after each workout on what they had based their exercise choice on, and these results showed that nine out of ten participants in the complete feedback group had typically used the list of suggestions for subsequent exercises (regularly combined with the body map, and often combined with their own preference), and only one participant typically based the choice on his/her own preference or on the body map alone. In line with these results, high scores were found for all items of the user experience questionnaire in the complete feedback group, and 100% of the participants indicated that they would like to use the feedback during their future workouts, confirming that this type of feedback was perceived to be useful and valuable. 

However, the feedback that consisted of the body map only, was not found to be significantly effective in improving participants’ cumulative muscle load balance. A potential explanation for this finding is that even when participants know which muscle groups they need to train (because these muscle groups appear white or light green on the body map), they do not know exactly which exercises to choose to target these muscle groups. Moreover, the questionnaires showed that 60% of the participants in the partial feedback group typically used the body map to select their next exercise, but 40% made the choice based on their own preference or another strategy. The fact that a substantial portion (40%) of the participants in the partial feedback group did not use the feedback could also be a reason for the lack of effect on the muscle load balance. To further examine this theory, we compared the average cumulative muscle load CVs between participants in the partial feedback group who did and did not use the body map. Interestingly, the mean cumulative muscle load CV was much smaller (76.8% of the mean cumulative muscle load) for the participants who did use the body map to select their next exercise, compared with participants who did not (97.1% of the mean cumulative muscle load). This finding highlights that feedback consisting of the body map alone could potentially aid in improving the cumulative muscle load balance, but not for all practitioners, either because some practitioners may prefer other methods to select their exercises or perhaps because they do not know how to properly use this type of feedback. In line with this, the user experience scores were lower in the partial feedback group than in the complete feedback group. This shows that the content of this feedback can be improved upon in terms of relevance and/or understandability [15]. 

The mean cumulative muscle loads increased more throughout the workouts for the control group compared with the feedback groups. This probably occurred because the feedback groups received feedback with a load level that was determined during the intake sessions and not adjusted during the relatively short (four-week) period of the study, and participants in the feedback groups seemed to adhere to these load levels. This result suggests that this type of feedback can be effective in guiding strength training practitioners towards a certain load level. Therewith, the application could potentially aid in preventing practitioners from training too heavily (and thereby presumably prevent overload injuries). The participants that did not receive feedback increased their loads, which could constitute a risk for overtraining according to the ACSM [19]. However, it must be noted that, especially when this feedback application is applied for longer training periods, practitioners are expected to increase in strength, implying that the load prescriptions should also increase over time (meaning we cannot exclude that the control group increased their load appropriately). It is very important that appropriate load increases are determined carefully. For instance, regular (submaximal) 1RM tests could be conducted to update the 1RM values in the application. Another option that may be more applicable, is that practitioners regularly evaluate if they can perform one or two repetitions more than the prescribed number, in which case the load prescriptions in the app could be increased with 2–10% according to ACSM guidelines [19]. 

Although similar studies regarding cumulative muscle load feedback are lacking in the literature, the present findings can be compared to those of studies that applied different forms of feedback during strength training, including feedback on barbell velocities and body kinematics and kinetics. In line with our findings, Weakley et al. [24] and Wilson et al. [14] found that visual barbell-velocity feedback improved motivation and competitiveness of strength training participants. Furthermore, Keller et al. [25] found that kinematic jump height feedback led to immediate improvement in jump performance. This agrees with the immediate improvement in muscle load balance for the complete feedback group observed in the first workout of the present study. However, when the feedback was removed in the study of Keller et al. [25], all improvements in jump performance were lost. Similarly, another feedback study found that the strength training performance returned to pre-feedback level when the feedback was removed [26]. Although this was not specifically investigated in the present study, the immediate improvement observed in the complete feedback group with respect to the control group at the first workout and absence of clear further improvements in the subsequent workouts may suggest there is no learning effect and people will depend on the application to obtain the positive effects. Strength training participants would therefore be advised to continue using the app if made available in practice.

Whereas typical methods to estimate muscle loads in human movement science studies include electromyography (EMG) or motion capture combined with musculoskeletal modeling [27], a new muscle load estimation method was chosen for the present study. A clear advantage of this new method compared to the more typical methods is that it can easily be applied in the gym, as no sensors or markers need to be attached to the athlete’s body. However, the validity of this new method has not been investigated yet, and it is expected that the muscle load estimations deviate from the actual muscle loads for multiple reasons. Firstly, the muscle load was estimated based on rough estimates of the muscle contributions per exercise, subdivided into either primary (100% contribution) or secondary (50% contribution) muscles. This subdivision in only two options is most likely an oversimplification, since muscles could also contribute with other percentages. Secondly, individual differences in anatomy and movement execution were not included in the estimation. It is important for future use of the application that the accuracy of the estimations is tested and further improved. Regarding the first point, the muscle force estimation could be improved if more detailed muscle force contributions are determined, for instance, by estimating the muscle forces with musculoskeletal modeling with or without EMG for multiple participants and multiple exercises, and using the mean estimated contributions to replace the current functional anatomy-based estimations. Regarding the second point, real-time measurements would be required to include individual anthropometry and movement executions in the muscle load estimation. Camera-based motion capture combined with a fast-computing musculoskeletal model is a recent development that may be a potential option to integrate within the feedback application in the future [28].

In addition to improving the accuracy of the muscle load calculations, other adjustments could be made to improve the applicability of the feedback application. In the present study, muscle load advice was always based on a full body workout, a fixed load percentage of 70% of the 1RM, and a fixed number of sets and repetitions. It would be valuable if the application was expanded with multiple training options, so that the muscle load advice could be adjusted based on the final training goal of the athlete (e.g., hypertrophy and speed). For instance, athletes should be able to choose the desired 1RM level, and whether to train a specific body part or the full body. Moreover, future research could focus on further optimizing the load advice based on personal characteristics including gender, age, and experience. 

The mean muscle soreness was not lower in the feedback groups than in the control group. However, the mean muscle soreness was found to be very low for all groups (with a score of 3 out of 100 on average). Moreover, muscle soreness cannot be translated to muscle overload injuries directly, and it is therefore important that the application is evaluated for a longer period wherein injuries are monitored prospectively as well to determine if the feedback can be effective for injury prevention. In addition, it must be noted that muscle soreness was self-assessed and based on subjective perceptions, and it may therefore be difficult to compare the amount of muscle soreness between individuals and between groups.

Multiple strengths and limitations apply to the present study. One of the strengths is that the studied feedback application can be directly applied in the sport practice and is currently already being used by Gymstory to provide muscle load feedback to their users. Another strength pertains to the innovative characteristics of the study, which, to our knowledge, is the first study to develop and evaluate a muscle load feedback application for strength training. On the other hand, the study suffered from some noteworthy limitations. Firstly, the sample size of the study was relatively small with 10 participants per group, especially because substantial differences between individuals were observed (as can be deducted from the substantial size of the error bars in Figure 3 and the difference in reported adherence within the partial feedback group). Secondly, recruitment was performed through voluntary response sampling, which may have led to self-selection bias. It is possible that the participants that volunteered were already more interested in receiving feedback during strength training than people that did not volunteer. Thirdly, we did not include a group who only received the exercise suggestion. Therefore, we cannot conclude whether the better muscle balance and user experience scores found for the complete feedback group were due to the combination of body map and exercise suggestion, or mainly due to the exercise suggestion. The absence of improvements in the partial feedback group may point at the second possibility.

## 5. Conclusions

Feedback regarding the personal muscle load of strength training participants provided in the form of a muscle body map and exercise suggestion can effectively aid in achieving a more balanced cumulative muscle load, which may decrease the risk of overloading certain muscles while underloading other muscles.Feedback regarding the muscle load in the form of a body map and suggested exercises was perceived to be valuable and stimulating.Feedback regarding the muscle load can effectively guide strength training participants towards a certain load level, presumably helping to maximize training effects without getting injured.Feedback regarding the muscle load does not change muscle soreness.Future research can improve the accuracy of the muscle load estimation.A longitudinal study with a longer follow-up period is needed to investigate if the feedback application can effectively prevent muscle injuries.

## 6. Practical Application

The proof-of-concept feedback application can successfully influence the cumulative muscle load of strength training practitioners. This highlights the potential of this type of feedback as a training tool. Future research should improve and validate the application and investigate if it can successfully prevent muscle overload injuries.

## Figures and Tables

**Figure 1 sports-11-00170-f001:**
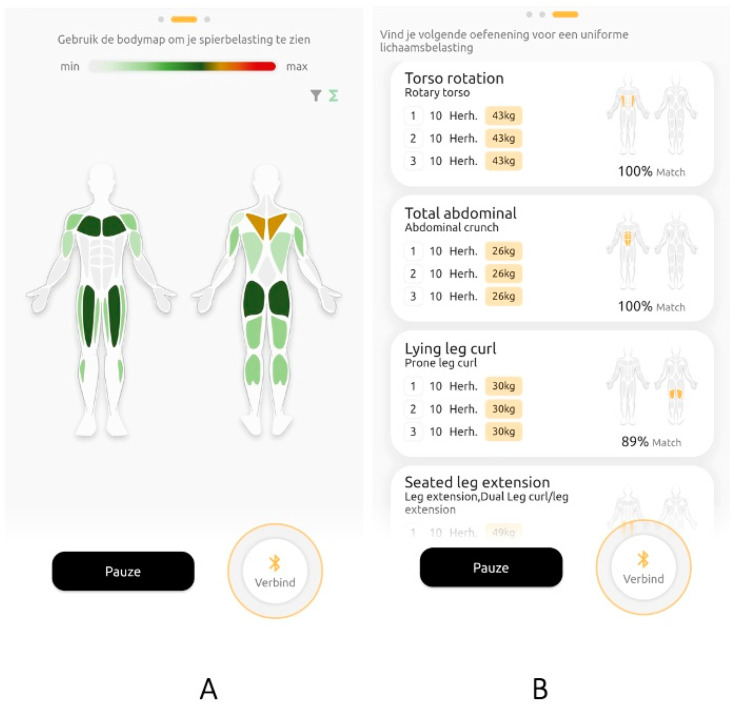
Screenshots of the Gymstory app with (**A**) the muscle load body map showing the cumulative muscle load per muscle by color and (**B**) a list with suggestions for subsequent exercises that target muscle groups that have not or had barely been loaded yet.

**Figure 2 sports-11-00170-f002:**
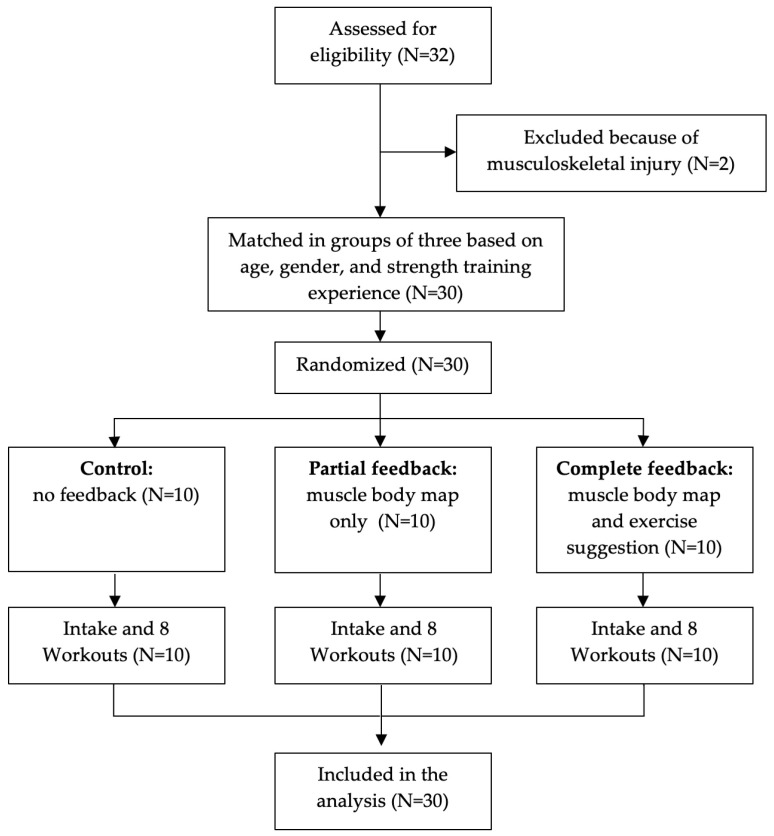
Flow diagram of the study design.

**Figure 3 sports-11-00170-f003:**
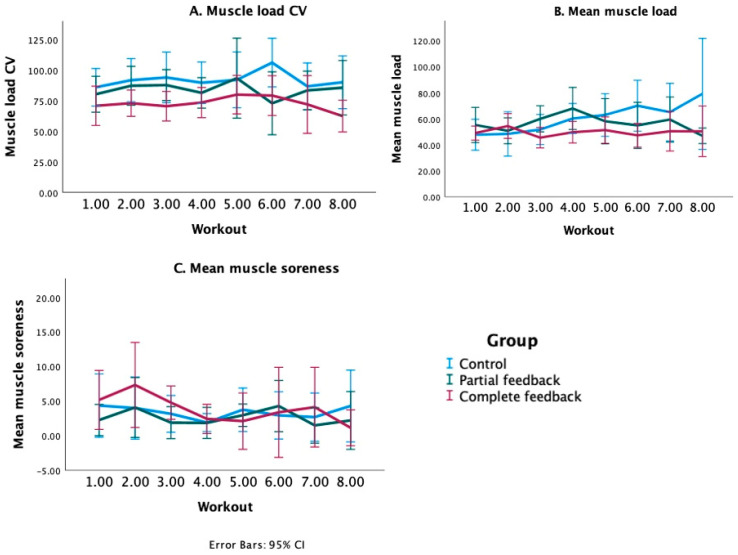
Group means of (**A**) the cumulative muscle load CV (coefficient of variation; indicator of muscle load balance), (**B**) the mean cumulative muscle load, (**C**) the mean muscle soreness, per workout. Error bars (95% confidence intervals (CI)) indicate the variation between participants.

**Figure 4 sports-11-00170-f004:**
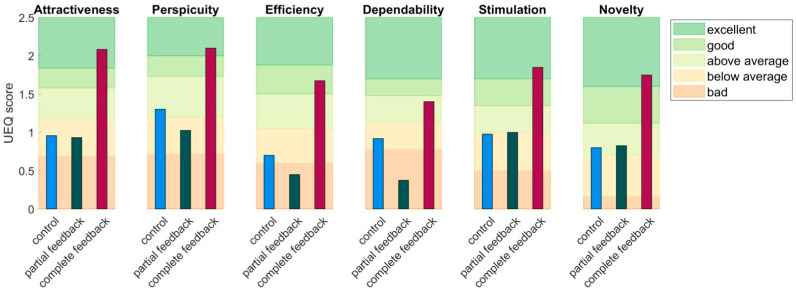
Overview of the mean scores on the different scales (subplots) for the different groups (colors) at the last training session. Shaded areas represent the benchmark values of <25%, 25–50%, 50–75%, 75–90%, and >90% performance, representing bad, below average, above average, good, and excellent performance, respectively.

**Table 1 sports-11-00170-t001:** Participant characteristics per group and the *p*-values of the one-way ANOVAs comparing those characteristics between the groups.

Item	Control (*n* = 10)		Partial Feedback (*n* = 10)		Complete Feedback (*n* = 10)		*p*-Value
	*n* (%)	Mean ± SD	*n* (%)	Mean ± SD	*n* (%)	Mean ± SD	
Sex Male Female	5 (50%) 5 (50%)		6 (60%) 4 (40%)		5 (50%) 5 (50%)		0.886
Age (years)		38 ± 14		39 ± 15		36 ± 15	0.934
Strength training experience (years)		2.5 ± 2.6		2.2 ± 2.4		2.4 ± 2.5	0.963

**Table 2 sports-11-00170-t002:** Parameter estimates (Beta) and their 95% confidence intervals (CIs) of the final Generalized Estimation Equation models for cumulative muscle load CV (coefficient of variation), mean cumulative muscle load, and mean muscle soreness.

Independent Variables		Cumulative Muscle Load CV (β and 95% CI)	Mean Cumulative Muscle Load (β and 95% CI)	Mean Muscle Soreness (β and 95% CI)
Group	Control ^a^	0	0	0
	Partial	−6.8 (−19.5, 6.0)	7.5 (−7.3, 22.3)	−0.5 (−2.8, 1.8)
	Complete	−18.9 (−29.3, −8.6) *	1.3 (−9.5, 12.0)	0.9 (−1.8, 3.6)
Workout number	1 ^a^	0	0	0
	2	5.0 (−4.4, 14.3)	0.7(−14.4, 15.7)	1.2 (−1.4, 3.8)
	3	5.1 (−3.9, 14.1)	4.0 (−7.8, 15.8)	−0.7 (−2.5, 1.2)
	4	2.5 (−5.0, 10.0)	12.5 (−4.8, 29.8)	−1.9 (−3.5,−0.2) *
	5	8.9 (−0.6, 18.4)	16.4 (−1.1, 33.9)	−1.0 (−2.4, 0.5)
	6	6.7 (−3.7, 17.1)	22.1 (0.9, 43.4) *	−0.3 (−2.1, 1.5)
	7	0.6 (−10.2, 11.3)	16.1(−0.9, 33.0)	−1.2 (−3.5, 1.0)
	8	−0.3 (−8.7, 8.1)	27.6 (4.7, 50.5) *	−1.1 (−3.5, 1.4)
Interaction group * workout number	Control*1 ^a^		0	
	Control*2 ^a^		0	
	Control*3 ^a^		0	
	Control*4 ^a^		0	
	Control*5 ^a^		0	
	Control*6 ^a^		0	
	Control*7 ^a^		0	
	Control*8 ^a^		0	
	Partial*1 ^a^		0	
	Partial*2		−5.2 (−24.0, 13.7)	
	Partial*3		0.7 (−15.5, 17.0)	
	Partial*4		0.2 (−19.4, 19.8)	
	Partial*5		−15.2 (−37.3, 7.0)	
	Partial*6		−24.0 (−47.9, −0.1) *	
	Partial*7		−13.5 (−33.9, 6.8)	
	Partial*8		−37.6 (−62.4, −12.9) *	
	Complete*1 ^a^		0	
	Complete*2		4.8 (−13.4, 22.9)	
	Complete*3		−7.6 (−21.3, 6.1)	
	Complete*4		−11.9(−30.1, 6.4)	
	Complete*5		−14.5 (−34.2, 5.3)	
	Complete*6		−24.3 (−46.3, −2.2) *	
	Complete*7		−14.3(−33.5, 4.9)	
	Complete*8		−26.3 (−49.5, −3.0) *	

* Refers to an interaction term. ^a^ Refers to a reference group.

**Table 3 sports-11-00170-t003:** Results of the one-way ANOVA comparing the user experience questionnaire (UEQ) scores on each scale between groups, as well as the post hoc analyses.

Scale	Post Hoc Comparisons	Mean Difference	Standard Error	df	*F*-Value	*p*-Value	Effect Size *r*
Attractiveness				2, 26	4.465	0.022 *	0.506
	Control–Partial	−0.080	0.430			1.000	
	Control–Complete	−1.127	0.418			0.036 *	
	Partial–Complete	−1.047	0.430			0.066	
Perspicuity				2, 26	3.826	0.035 *	0.477
	Control–Partial	0.161	0.377			1.000	
	Control–Complete	−0.800	0.367			0.116	
	Partial–Complete	−0.961	0.377			0.051	
Efficiency				2, 26	3.810	0.035 *	0.476
	Control–Partial	0.200	0.462			1.000	
	Control–Complete	−0.975	0.449			0.118	
	Partial–Complete	−1.175	0.462			0.051	
Dependability				2, 26	4.403	0.023 *	0.503
	Control–Partial	0.500	0.332			0.430	
	Control–Complete	−0.483	0.323			0.439	
	Partial–Complete	−0.983	0.331			0.019 *	
Stimulation				2, 26	4.402	0.023 *	0.503
	Control–Partial	−0.136	0.325			1.000	
	Control–Complete	−0.875	0.316			0.031 *	
	Partial–Complete	−0.739	0.325			0.094	
Novelty				2, 26	5.170	0.013 *	0.533
	Control–Partial	−0.117	0.329			1.000	
	Control–Complete	−0.950	0.320			0.019 *	
	Partial–Complete	−0.833	0.329			0.053	

***** *p* < 0.05.

## Data Availability

The data presented in this study are available on request from the corresponding author. The data are not publicly available due privacy reasons.

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
