# Peer review of "A Muscle Load Feedback Application for Strength Training: A Proof-of-Concept Study"

_sports, 2023, doi:10.3390/sports11090170_

Round 1

Reviewer 1 Report

This study investigates whether muscle load feedback can improve balance and muscle soreness in strength training practitioners. It also looks at the user experience and motivational effect of the muscle load feedback application.

Lines 30-33: The opening two sentences are contradictory. You need another sentence between them as a link to clarify your meaning. Something like "However, despite an increase in participation in exercise participation, sedentary behaviours remain a serious health issue..."

Table 2 might be better split. Keep the Group data at the top in the table and put the Workout number data in a figure to match Figure 2A, B and C.

Line 327, 335, 374: Change "compared to" to "compared with"

Line 387: How can a 1RM test be submaximal?

The observation that muscle soreness was low in these participants is not surprising since they are all experienced lifters. Perhaps running this experiment with novice lifters would provide more insight.

Reviewer 2 Report

Many thanks to the editors for having me to review this work and to the authors for their time dedicated to this interesting paper. The following is a series of revisions with the intention of contributing to the paper.

Starting with the abstract, it should be pointed out that the title of the abstract is missing. The objective of the abstract is not clearly identified. In addition, the measurement instruments should be indicated.

Regarding the introduction, the review of the most recent studies is good, although they should focus on the studies that are most directly related to their variables and select those that have been carried out with a larger population and are more recent. In the introduction, the objective can be seen in a concrete way.

Regarding the methodology, in the section on participants they should indicate the type of sampling. The procedure is very well explained and it is a novel methodology with the use of APP. The validation process of this app as a measurement tool should be further emphasized. In the statistical analysis section, it should be noted that frequency analysis has been performed.

In the results, it is recommended that Figure 3 be shown more clearly.

Regarding the discussion, it is considered that the presence of other studies with which to compare should be greater. There is no coherence between a good review in the introduction and the absence of studies in the discussion; the authors have material with which to compare, although this is why they have to modify the discussion of the work so that they compare and interpret the results in the discussion.

As for the conclusions, there are too many conclusions with respect to the objective, although there is more than enough data to draw the conclusions. Perhaps the authors should consider some secondary objectives more concretely.

Thank you

Reviewer 3 Report

Model for Manuscript Review

Research articles

Manuscript title:

Sports-2519029- A muscle load feedback application for strength training: a proof-of-concept study

Títle

Is it understandable and concise?

( x ) Yes (  ) Not

Reflects the content?

( x ) Yes (  ) Not

Abstract

It includes: objectives, methodology, key findings and conclusions?

(  ) Yes ( x ) Not

Introduccion

The investigation was carried out in a suitable theoretical structure?

(  ) Yes ( x ) Not

Clear leaves the questions you want to answer and objectives of the work?

(  ) Yes ( x ) Not

The cited references are current and relevant?

(  ) Yes ( x ) Not

Methods

The methods presented are appropriate to achieve the proposed objectives?

(  ) Yes ( x ) Not

The selection and composition of the sample are adequately described?

(  ) Yes ( x ) Not

The data collection process and the tools used are described clearly?

(  ) Yes ( x ) Not

The statistical analysis and the research design appropriate?

(  ) Yes ( x ) Not

Results

The presentation of the results clear?

(  ) Yes ( x ) Not

The main results are highlighted without the inclusion of interpretation and comparisons?

(  ) Yes ( x ) Not

The results evaluate the proposed objectives?

(  ) Yes ( x ) Not

Tables and figures are properly numbered, labeled and explained?

(  ) Yes ( x ) Not

Discussion and Conclusion

The results are discussed based on the literature?

(  ) Yes ( x ) Not

Author's interpretations show the safety and soundness?

(  ) Yes ( x ) Not

The limitations of the work are presented?

( x ) Yes (  ) Not

The conclusions of the study are presented?

(  ) Yes ( x ) Not

The conclusions respond to the objectives?

(  ) Yes ( x ) Not

General comments:

Title

Are presented satisfactorily.

Abstract

The Abstract does not follow the rules of the journal, not present itself in a structured way.

Although the manuscript tries to apply a pallicative, there is no better result or description of what was done to understand the study in a more abstract way. It is suggested that the methodology be better detailed and that the statistical and absolute results be presented.

In conclusion, we indicate that practical applications should be presented

Please confirm if the keywords match descriptors in health sciences.

Introduction

The first paragraph starts with bodybuilding or strength training in the abstract, not addressing the use of this training specifically. Continuing, it addresses strength training focused on weightlifting, powerlifting and cross fit. There is a relationship between the three types of training. In the view of the authors, the trainings have the same objectives, are done in the same way and use the same components of the load and the same principles of the sports training. Please clarify.

Despite the manuscript proposing the use of an application, in the introduction there was no concern with the construct that one wants to validate. The introduction should cover the construct that justifies the use and validation of the application, which does not happen.

The last paragraph, the problem is not properly identified. It would be advisable to place studies for and against that demonstrate what was mentioned in the previous paragraphs and then, there would be support for the proposed objectives.

At the end of the last paragraph of the introduction, the study hypotheses will be indicated.

In view of the above, a question arises, why topics such as overtraining and overreaching were not addressed, since an inappropriate recovery could lead to a greater risk of injuries.

The introduction should be completely rewritten and the theme in question, the use of a tool should be better approached and built, and its application should be better explained since strength training is very comprehensive.

Methods

It should present the design of the study. A CONSORT or timeline should be presented in order to get a better view of the study design.

The sample should be better explained with the number of subjects presented initially and then present the inclusion and exclusion criteria.

How was the sample size determined? Was any statistical program or formula used to support the number? Please explain.

In addition, men and women were adopted as samples. Should fatigue, load components, and strength training principles be approached the same way for men and women? Please explain.

Instruments must be referenced correctly. In the case of instruments, it would be feasible to present the model in parentheses, manufacturers, city, state, and country of manufacture. In the case of questionnaires, present the references of who created it, who validated it for the language in which the study was carried out, and the cutoff points. It would also be indicated that a summary of the questionnaire was presented, if applicable.

Figure 1 must be correctly formatted, that is, the title of the figure comes after it. There is no clearer and more objective description of the program presented, there is no observance of criteria for the validation of the instrument.

Procedures should be better explained. Furthermore, I suggest that the authors consult the validation manuscripts and adopt the procedures described in these manuscripts, such as the need to test and retest to confirm the ANOVA. Verification, through a well-written constriction, if the instrument really verifies what it proposes, and also, if possible, confirmation through the use of an already validated instrument, since there are several instruments in use today in the market and which are considered the gold standard.

In addition, broader statistical measures would be necessary to confirm that the proposed instrument is really valid. He also suggests that they subsequently do a Bland Altman test to justify the findings in terms of proper disposal.

Statistical treatment should be better detailed in order to better follow what has been done. No mention was made of which measures of central tendencies were used, the test for normality, and the effect size. Correlation cutoff points were not referenced, and descriptive measures were used in validation manuscripts, in addition to data confirmation through comparison with other instruments. Suggest you consult Cohen J. Statistical Power Analysis for the Behavioral Sciences (2nd ed.), New Jersey: Lawrence Erlbaum Associates, 1988.

Results

The results should bring the representation, through tables and figures of everything mentioned in the methodology, which does not occur. I suggest completing the presentation of results, emphasizing what was mentioned in the review on statistical treatment.

It suggests that the aforementioned be observed so that the study can effectively validate the proposal. Many of the results presented must be confronted and compared with the use of other instruments, such as:

How were muscle load values, cumulative work, and muscle soreness calculated? Data were compared. For example, with the use of an algometer to check muscle pain. Was there any comparison with other blood biochemical indicators to see if the so-called muscle load really indicates this?

From what was determined what would be Muscle soreness and what would be User experience? We were not able to visualize the construct for this in the manuscript, let alone some type of evaluation, with the use of other instruments that allow us to affirm what is being affirmed.

The results must be based on the introduction, response to the objectives, and the objectives must determine the methodology. The methodology must allow the replication of the study. The results, on the other hand, must provide answers to be discussed based on the objectives. It doesn't happen.

Discussion

It should reaffirm the objectives and start discussing the results in the chronological order that appears in the item results.

Despite being relatively well presented, the discussion does not try to explain the study outcomes, limiting itself only and presenting studies in favor and against the findings.

The discussion, for the number of findings, was very small and did not exhaust the discussion around the proposal. Suggest that the discussion be addressed in greater depth, perhaps a couple more paragraphs would help to support the presented conclusions.

We suggest that the discussion be carried out in the chronological order in which the results were presented. In the discussion, many of the concepts and definition could be better explored so that the instrument was better understood and the construct as well.

The limitations mention the size of the groups, but the results do not indicate that this number of people has the power to justify the conclusions, for example.

Conclusion

Are presented satisfactorily. However, the conclusions address issues that were not the subject of the study.

It also suggests that practical applications of the findings are presented.

Mention the practical applications in relation to the results found.

References

Of the 26 references, only 10 are current, 15 are more than five years old, and one doesn't even have a year. Please review the formatting of the references so that they comply with the journal's standards and update the manuscript's theoretical framework.

Overview

The manuscript presented addresses a relevant research topic.

It would be advisable to do a general review.

Some points end up being unclear, for example, the program was created from what? What could be answered in the text if the construct of the proposed instrument had been mentioned.

Specific comments and suggestions:

Outcome evaluation

  • Accept unchanged                       (   )
  • Accepted with minor changes       (   )
  • Accepted with major changes       ( x )

·       Rejected                                      (   )

Round 2

Reviewer 3 Report

After the adaptations presented, I consider the manuscript in conditions to be published